# Understanding COVID-19 vaccination behaviors and intentions in Ghana: A Behavioral Insights (BI) study

Swathi Vepachedu[1,2]*, Anastasiia Nurzenska[3], Anna-Leena Lohiniva[3], Al-hassan Hudi[3], Sena Deku[4], Julianne Birungi[5], Karen Greiner[5], Joseph Sherlock[1], Chelsi Campbell[1,2], Lori Foster[2,6]

1 Center for Advanced Hindsight, Duke University, Durham, NC, United States of America, 2 Department of Psychology, North Carolina State University, Raleigh, NC, United States of America, 3 UNICEF Ghana Country Office, Accra, Ghana, 4 VIAMO, Accra, Ghana, 5 UNICEF Regional Office Central and West Africa, Dakar, Senegal, 6 School of Management Studies, University of Cape Town, Cape Town, South Africa

* svepach@ncsu.edu

## Abstract

### Introduction

Vaccine uptake is influenced by a variety of factors. Behavioral Insights (BI) can be used to address vaccine hesitancy to understand the factors that influence the decision to take or refuse a vaccine.

### Methodology

This two-part study consisted of a survey designed to identify the influence of various drivers of people's COVID-19 vaccination status and their intention to take the vaccine in Ghana, as well as an experiment to test which of several behaviorally informed message frames had the greatest effect on vaccine acceptance. Data was collected from a total of 1494 participants; 1089 respondents (73%) reported already being vaccinated and 405 respondents (27%) reported not being vaccinated yet. The mobile phone-based surveys were conducted between December 2021 and January 2022 using Random Digit Dialing (RDD) to recruit study participants. Data analysis included regression models, relative weights analyses, and ANOVAs.

### Results

The findings indicated that vaccine uptake in Ghana is influenced more by social factors (what others think) than by practical factors such as ease of vaccination. Respondents' perceptions of their family's and religious leaders' attitudes towards the vaccine were among the most influential drivers. Unexpectedly, healthcare providers' positive attitudes about the COVID-19 vaccine had a significant negative relationship with respondents' vaccination behavior. Vaccine intention was positively predicted by risk perception, ease of vaccination, and the degree to which respondents considered the vaccine effective. Perceptions of religious leaders' attitudes also significantly and positively predicted respondents' intention to

**Data Availability Statement:** All relevant data are within the paper and its Supporting Information files.

**Funding:** The authors received no specific funding for this work.

**Competing interests:** The authors have declared that no competing interests exist.

get vaccinated. Although perceptions of religious leaders' views about the vaccine are an important driver of vaccine acceptance, results asking respondents to rank-order who influences them suggest that people may not be consciously aware—or do not want to admit—the degree to which they are affected by what religious leaders think. Message frames that included fear, altruism, social norms were all followed by positive responses toward the vaccine, as were messages with three distinct messengers: Ghana Health Services, a doctor, and religious leaders.

## Conclusions

What drives COVID-19 vaccine intentions does not necessarily drive behaviors. The results of this study can be used to develop appropriate COVID-19 vaccine uptake strategies targeting the most important drivers of COVID-19 vaccine acceptance, using effective message frames.

## Introduction

Vaccines have long been considered one of the most important public health interventions in the history of modern medicine, resulting in a decline in morbidity and mortality of various infectious diseases worldwide [1]. Sufficient uptake of vaccines typically results in herd immunity, which is required to ensure an infectious agent is no longer able to spread. However, vaccine hesitancy, when people are hesitant to take a vaccine, has always been an issue. This is particularly common when new vaccines are introduced [2]. Vaccine hesitancy is a global phenomenon that the WHO identified as one of the top ten global health threats in 2019 [3]. Countries have been experiencing varying levels of COVID-19 vaccine hesitancy across the globe [3–6]. A publicly available behavioral tracker of COVID-19 demonstrates that vaccine hesitancy has fluctuated during the pandemic depending on epidemiological and other contextual factors [7], which aligns with research findings that vaccine hesitancy is related to context and is time bound [8]. Studies in Ghana show that a significant percentage of the adult population are hesitant to take the COVID-19 vaccine. A cross-sectional survey conducted in Ghana at the end of 2020 indicated that approximately 65% of the participants were willing to take the vaccine, however, another study conducted in early 2021 indicated that only approximately half of the respondents were willing to take the vaccine, that just over one-fifth (21%) of the respondents were unlikely to take the vaccine, and another 28% were undecided [9]. Hesitancy has been also identified among healthcare providers in Ghana [8, 10].

Vaccine Hesitancy refers to delay in acceptance or refusal of vaccination despite availability of vaccination services [11]. It is a complex and context-specific phenomenon, varying across time, place and vaccine type [8, 11]. Therefore, it is difficult to predict exactly how vaccines will be received in any given setting [12]. Hesitancy can be influenced by environmental factors such as physical availability, affordability, willingness to pay, geographical accessibility, ability to understand (language and health literacy), and the ability of immunization services to provide vaccines. It can also be influenced by cultural, social and behavioral factors, including trust in the effectiveness and safety of vaccines, in the system that delivers them, in the reliability and competence of health services and health professionals, and the motivations of policymakers who decide on the needed vaccines in the vaccination program [8, 13]. A growing number of studies have identified demographic and socioeconomic factors linked with vaccine acceptance such as age or marital status [14, 15].

Rapid systematic reviews have identified varying factors influencing COVID-19 vaccine hesitancy, including perceived risk, concerns over vaccine safety and effectiveness, doctors' recommendations, inoculation history [5], low levels of education and awareness, inefficient government efforts and initiatives, poor influenza-vaccination history, as well as conspiracy theories relating to infertility about the COVID-19 vaccine [16]. Generally, COVID-19 vaccine hesitancy has been further fueled by conspiracy theories, especially through social media channels [17, 18]. Studies exploring factors that influence COVID-19 vaccine hesitancy in Ghana are limited. One online survey study among adults in Ghana explored the relationship between personal health engagement, fear of COVID19 and COVID19 susceptivity, and its impact to vaccine attitude and vaccine willingness. The study concluded that fear influences vaccine attitude and through vaccine willingness highlighting the importance of fear-based messaging when promoting COVID-19 vaccine in Ghana [19]. Another cross- sectional online study conducted in Ghana explored vaccine hesitancy attitudes and identified the likelihood of participation or non-participation in the COVID-19 vaccination response. The study concluded that perceived benefits in both individual level and in the population level as well as reduced misinformation were likely to increase the participation [9]. In addition, a mixed methods study that looked into the impact of trust towards and other factors to the willingness of community members and community leaders to take COVID-19 vaccine concluded that mistrust towards political actors, among other things such as belief in superior protection of God, and misunderstandings about the vaccine development process negatively influenced willingness to take the COVID-19 vaccine [20].

In Ghana, the first phase of the COVID-19 vaccination campaign ran in March 2021 during which 3.3 million doses of vaccines were administered. Since then, COVID-19 vaccination campaigns have been initiated whenever vaccines have been made available. Ghana Health Services (GHS) has used multiple vaccine demand creation strategies including community outreach through community leaders, social media and mass media. They have also effectively responded to circulating rumors and debunked misinformation through a misinformation taskforce that monitors both online and offline media [21]. As of 31st July 2022, as much as 18,396,070 doses of COVID-19 vaccines were administered, constituting 35.5% of the total population fully vaccinated and 25.6% having taken at least 1 dose of the vaccine. Ghana Health Services estimates that by December 2022, 22.9 million adults in Ghana will have received the COVID-19 vaccine.

Behavioral Insights (BI), which involves the study of human behaviour, often drawing on empirical research in fields including economics, psychology and sociology and the use of these insights to design and make discrete changes in the environment to impact behaviours [22]. It can be used to address vaccine hesitancy to understand the factors that influence decisions to take or refuse a vaccine. The starting point of BI is understanding that humans are not always fully consciously aware of what affects their decisions. Decision-making often includes unconscious motivations, judgements, and decisions such as various heuristics and mental shortcuts [23–25]. Instead of asking people directly what influences their behavior. For example, BI uncover behavioral drivers by experiments and surveys using behavioral analytics to identify the relative influence and variety of factors that draw people to or away from vaccinations [26–28]. BI can also create strategies to improve decision-making, called nudges, which alter how choices are presented, leading decision-makers to behave in predictable ways. The key is to develop nudges in which the information is framed in a way that influences decision-making. COVID-19 message frame testing shows that different message frames work better in different environments, confirming the need for context specificity and highlighting the importance of identifying the appropriate frame for each context [29–31]. In addition, BI can encompass strategies that, unlike nudges, require sustained effort. For example,

implementation scientists have long recognized that social motivation, incentives, and rewards are crucial levers of behavior change [32–34].

This study used BI to better understand factors that influence COVID-19 uptake in Ghana and tested the impact of differently framed nudges on the willingness of people to take the vaccine. The study also differentiates between intentions and behavior as intention does not always lead to behavior change [35]. The findings of the study will be used to support vaccination demand generation in Ghana.

## Methodology

This study consisted of two components, which were both embedded in a survey administered to respondents from Ghana between December 2021 and January 2022. COVID-19 vaccines, including the AstraZeneca/Oxford and Pfizer-BioNTech vaccines, were available from February 24, 2021 [36]. By the time of data collection for this study, nearly 10% of the population was fully vaccinated and 22% received at least one dose of the vaccine.

The first part of the survey was designed to identify the influence of various drivers of vaccine acceptance. The second part of the survey included an experiment to test which of several behaviorally informed message frames had the greatest effect on vaccine acceptance. Both studies align with the Define, Diagnose, Design, and Test (DDDT) methodology used in BI [37] and were developed in collaboration with UNICEF.

### Participants

Data was collected from 2,088 mobile survey respondents in Ghana. Respondents were required to provide their consent and be at least 18 years old to participate in the survey. The sample size was determined by comparing 5 experimental treatments to a shared control arm. Using data from similar experiments [38], our computation was based on 80% power to detect a difference in means of 0.4 (M1 = 2.9 M2 = 2.5) for any two arms, assuming a standard deviation (SD) of 1.5 and 2.2 for means respectively on a 5-point Likert scale with a 2-sided significance level of 0.05. Inserting these into STATA's sample size calculation function (sampsi 2.9 2.5, SD1(1.5) SD2(2.2) power (.80), a sample of 348 for each arm was estimated. The total required sample with six experimental conditions was 2,088.

### Data cleaning

In cleaning the data, emphasis was placed on securing high data quality to achieve the goals of the study. A key benchmark was to focus on data from respondents with fully completed survey interviews. Given that with mobile based surveys, there was a possibility of a respondent going through the survey multiple times on different calls, duplicate and incomplete responses from unique respondents were also eliminated. Using this criterion, we excluded data from 594 respondents, resulting in a final sample of 1,494 respondents. All the results reported in this manuscript are from the overall sample with 1,494 respondents or a sub-sample of the same. Of the 1,494 respondents, the majority of the respondents were male (65%) and from rural areas of Ghana (63%). Most of the respondents were fairly young, with 84% between the ages of 18 and 35 years. Most of the respondents were not health-care workers (80%).

### Data collection

Data was collected in partnership with Viamo, a digital technology platform that provides interactive and measurable mobile engagement surveys. Viamo conducts mobile surveys and

gathers data from hard-to-reach communities. Random Digit Dialing (RDD) was used to recruit study respondents, a method wherein randomly generated mobile numbers are called.

Every member of the target *national* population had an equal chance of selection. The survey was delivered by Viamo in December 2021 and all data was collected by January 2022. Viamo administered the survey via Interactive Voice Response (IVR), an automated system that delivers pre-recorded voice messages to people over a mobile phone. The IVR system allows voice messages to be sent to users and for users to send messages back to the IVR system. Initial calls were placed between 8am and 8pm local time. Respondents who missed the call or were unable or unwilling to complete the survey at the time of the call could call back using the number recorded in their call logs or by using the redial feature to take the survey at a more convenient time. Each number was dialed only once. Calls were made available in six generally spoken languages in Ghana: English, Hausa, Twi, Ewe, Ga, and Dagbani. Respondents could select the preferred survey language by pressing a number on their telephone keypad. Any person who answered the phone was eligible for survey participation if they were at least 18 years old. The only caveat for mobile surveys was that the survey participant had to be in possession of an active mobile phone. This was mainly possible due to the high mobile phone penetration in the country.

## Ethical considerations

The study protocol was approved by the Ethical Board of Ghana Health Services. The study is voluntary and accordingly there is no penalty for refusing to participate. The study design allows participants to skip questions. The study was also confidential. Only a unique subscriber identifier was used for each contact number on the mobile survey platform.

The user contact information was not given out, with a unique identifier generated and used for each participating user. Informed consent was administered before the start of the survey. Upon answering the call, participants listened to a brief description of the study, including the duration of the survey. They were also informed about data confidentiality and were then requested to provide consent if they wished to proceed with the survey by pressing a number on their phones. Only after a participant's consent and age was recorded, the participant was able to proceed with the survey.

## Design

**Part I.** In Part I of the study, eight predictors were examined. These predictors measured various drivers of vaccination behavior and intentions, wherein respondents rated their perceptions on each of these drivers. The eight predictors included risk perceptions about the COVID-19 vaccine, effectiveness of the COVID-19 vaccine, ease of vaccination, and respondents' own attitudes toward the COVID-19 vaccine as well as their perceptions of their family's attitudes, community's attitudes, religious leaders' attitudes, and healthcare providers' attitudes toward the COVID-19 vaccine.

Two criterion variables were measured in Part I of the study: vaccination behavior and vaccination intention. First, we measured actual vaccination behavior–that is, whether respondents reported already receiving one or more doses of the COVID-19 vaccine. Second, for those who did not report receiving one or more doses of the COVID-19 vaccine, we measured their intention to get vaccinated.

**Part II.** The second part of the survey included one independent variable, *BI message types*, with six conditions. The BI message types were recorded by voice actors and were presented in the language chosen by the participant at the beginning of the survey. Respondents were randomly assigned to one of six possible messages, each of which included a BI element pertaining

to the framing of the message or the messenger. The six conditions were fear framing, altruism framing, social norms framing, Ghana Health Services as the messenger, a doctor as the messenger, and religious leaders as messengers. See Appendix A in S1 Appendix for a list of the six BI message types and their content included in the study.

Three dependent variables were examined in Part II of the study. First, respondents' willingness to recommend the vaccine to family and friends was measured. Second, their willingness to share the benefits of vaccination with their family and friends was measured. Lastly, respondents' intention to get vaccinated was measured again after presenting the BI message.

## Procedure

Upon receiving and answering the automated call to participate in the survey, respondents (n = 1494) were asked to choose among the six predominantly spoken languages in Ghana: English, Twi, Ga, Ewe, Dagbani and Hausa. Translations for each language were double-checked with key language experts to ascertain and confirm equivalence across languages for the survey. The rest of the survey was presented in the selected language. Respondents were then asked to provide their voluntary consent to participate in the survey and to verify their age. The survey immediately ended for respondents who did not consent to the survey and/or who reported being under 18 years. Throughout the survey, respondents could not go back to previous questions.

In part I of the study, respondents (n = 1494) were first asked about general vaccination behavior and if they ever received a vaccine as a child or an adult. They were then asked if they had received one or more doses of the COVID-19 vaccine. If the respondents (n = 405) reported not receiving one or more doses of the COVID-19 vaccine, they were presented with three questions measuring their intentions to get vaccinated against COVID-19. All respondents (n = 1494) then rated their perceptions on the various drivers of vaccination, namely, risk perceptions, effectiveness, ease of vaccination, and respondents' own attitudes toward the COVID-19 vaccine as well as their perceptions of their family's attitudes, community's attitudes, religious leaders' attitudes, and healthcare providers' attitudes toward the COVID-19 vaccine. After rating the prospective drivers of vaccination, respondents (n = 1494) were given a list of possible drivers and asked to indicate what they believe to be the most important and the least important influences on their decision to get vaccinated.

Next, respondents were asked to provide demographic information, including, gender, education, location (urban/rural), whether they had children, whether they were healthcare workers, whether they or their immediate family were infected with COVID-19, religious affiliation, medical status, and medical preference.

Part II of the study was presented after the demographic questions. Respondents were randomly assigned to one of six BI message types (see Appendix A in S1 Appendix). These were presented in the language selected by the participant at the beginning of the survey. After listening, respondents were asked if they would like to hear the BI message one more time and if they understood the message. Next, all respondents (n = 1494), regardless of vaccination status, were asked if they would recommend getting vaccinated to a family member or friend and if they would share the benefits of vaccination with a family member or friend. Unvaccinated respondents (n = 405) were then once again presented with the three questions measuring their intentions to get vaccinated, which were also presented during Part I of the study. Upon completing the survey, respondents were thanked and given an airtime reward of GHS10. This correlated to about USD $1.65 at the time of the study.

## Measures

Vaccination behavior was measured by asking respondents to use a binary Yes/No response scale to indicate whether they had received one or more doses of the COVID-19 vaccine. The

item included to measure vaccination behavior was, "Have you personally received one or more doses of the COVID-19 vaccine?"

Vaccination intention measured respondents' intention to get vaccinated against COVID-19 if they indicated not receiving one or more doses of the vaccine. Three items were used to measure vaccination intention. Respondents' ratings on the three items were averaged to form a composite which was used in all the analyses (α = 0.89). All three items were administered on a five-point Likert scale ranging from 1 (extremely unlikely) to 5 (extremely likely). An example item is, "How likely are you to consider getting vaccinated once the COVID-19 vaccine becomes available?"

Willingness to recommend the COVID-19 vaccine to a family member or friend was measured with a single item, which asked respondents: "When the vaccine becomes available, will you recommend the COVID-19 vaccine to a family member or friend?" This question was presented to respondents after they listened to the BI message (jingle). The three response options were yes, no, and don't know.

Willingness to share the benefits of vaccination to a family member or friend was measured with a single item, which asked: "Will you try to convince a family member or friend to get vaccinated against COVID-19?" This question was also presented after respondents listened to the BI message. The three response options were yes, no, and don't know.

Risk perceptions (α = 0.84, ω = 0.84) measured the degree to which respondents sensed a risk of contracting COVID-19. Two items were included to measure risk perceptions using a five-point Likert scale ranging from 1 (extremely unlikely) to 5 (extremely likely). An example item is, "If you do not get the vaccine, how likely do you think it is that you will catch COVID-19?"

Effectiveness (α = 0.79, ω = 0.79), adapted from a previous study [39], measured respondents' confidence in the COVID-19 vaccine. Two items were included using a five-point Likert scale ranging from 1 (extremely unsafe) to 5 (extremely safe) for the first item and a five-point Likert scale ranging from 1 (extremely ineffective) to 5 (extremely effective) for the second item. An example item is, "How safe do you think the COVID-19 vaccine is?"

Ease of vaccination was measured using two items for those who were vaccinated (α = 0.92, ω = 0.92) and two parallel items (α = 0.97, ω = 0.97) were included for those who were not vaccinated. All items used a five-point Likert scale. The first item asked directly about ease and used a scale ranging from 1 (very difficult) to 5 (very easy). The second item asked about the clarity of the process for getting vaccinated and used a scale ranging from 1 (very unclear) to 5 (very clear). An example item for those who reported being vaccinated is, "How easy was it for you to get the vaccine?" and for those who reported not being vaccinated is, "How easy will it be for you to get the vaccine?"

Own attitudes (α = 0.88, ω = 0.88) were measured using the following two items which assessed respondents' own attitudes toward the COVID-19 vaccine. The first item, "What is your attitude toward the COVID-19 vaccine generally?" was paired with a five-point Likert-type response scale ranging from 1 (very negative) to 5 (very positive); this response scale was also used to measure perceptions of others' attitudes (i.e., family, community, religious leaders and healthcare providers) The second item, "How important do you think it is to get vaccinated?" was paired with a five-point Likert-type scale ranging from 1 (not at all important) to 5 (extremely important), which was subsequently used to measure respondents' perceptions of others' attitudes toward the COVID-19 vaccine, described next.

Family's attitudes (α = 0.87, ω = 0.87) were measured using the following two items which assessed respondents' perceptions of their family's attitudes toward the COVID-19 vaccine: "What is your family's attitude toward the COVID-19 vaccine?" and "How important does your family think it is for you to get the vaccine?"

Community's attitudes (α = 0.86, ω = 0.86) were measured using the following two items which assessed respondents' perceptions of their community members' attitudes toward the COVID-19 vaccine: "What is your community's attitudes toward the COVID-19 vaccine?" and "How important does your community think it is for you to get the vaccine?"

Religious leaders' attitudes (α = 0.88, ω = 0.88) were measured using the following two items which assessed respondents' perceptions of religious leaders' attitudes toward the COVID-19 vaccine: "What are your religious leaders' attitudes toward the COVID-19 vaccine?" and "How important do your religious leaders think it is for you to get the vaccine?"

Healthcare providers' attitudes (α = 0.86, ω = 0.86) were measured using the following two items which assessed respondents' perceptions of healthcare providers' attitudes toward the COVID-19 vaccine: "What are your healthcare providers' attitudes toward the COVID-19 vaccine?" and "How important do your healthcare providers think it is for you to get the vaccine?"

Correlations between each of the drivers for both unvaccinated and vaccinated participants are included in Appendices A and B in S1 Appendix.

Self-ranked top influences measured who respondents consciously perceived to have the biggest influence over their decision to get vaccinated. This was assessed using a single item: "Of these choices, who has the biggest influence over your decision about whether to get the COVID-19 vaccine?" Respondents were presented with the following options: self (own attitudes), family, community members, Ghana Health Service, religious leaders, and healthcare providers.

Self-ranked least influences measured who respondents consciously perceived to have the least influence over their decision to get vaccinated. It was measured using a single item: "Who has the least amount of influence over your decision about whether to get the COVID-19 vaccine?" Respondents were presented with the same choices indicated above: self (own attitudes), family, community members, Ghana Health Service, religious leaders, and healthcare providers.

## Results

Table 1 summarizes the key findings for each of the research questions studied. Specifically, results from the relative weights analysis are summarized identifying the highest and lowest contributing drivers for people's vaccination behavior and intention. Further, results of people's self-ranked influences on their decision to get the vaccine are also summarized below along with the effects of the BI message types on individuals' willingness to recommend the vaccine and share its benefits with others.

### Part I—Vaccination behavior

Self-reported vaccination behavior was examined for the overall sample and disaggregated by sociodemographic characteristics of the sample (Table 2). As can be seen, 73% of the sample reported receiving at least one dose of the COVID-19 vaccine. Subgroup vaccination rates ranged from a low of 64% for women to a high of 86% for healthcare workers.

Next, the sample was restricted to those (N = 1089) who reported receiving at least one dose of the COVID-19 vaccine. Descriptive statistics were examined for each of the eight potential drivers (Table 3). As can be seen by the average ratings reported in Table 3, respondents perceived themselves and healthcare providers to have relatively positive attitudes toward the COVID-19 vaccine, with average ratings at 4.17 and 4.21 respectively on the 5-point scale. Other perceptions were a bit less favorable. For example, while still above the midpoint, ratings of ease of vaccination were lower, with an average rating of 3.66, suggesting that there was

**Table 1. Summary of key findings.**

| Research Questions | Highest contributing drivers | Lowest contributing drivers | Key insights |
|---|---|---|---|
| **1: What drives people's vaccination behavior?** | i. Family's attitudes (18.6%)<br>ii. Own attitudes (17.5%)<br>iii. Religious leaders' attitudes (16.7%)<br>iv. Community's attitudes (16.5%) | i. Effectiveness (9.7%)<br>ii. Healthcare providers' attitudes (8.6%)<br>iii. Risk Perception (5.7%)<br>iv. Ease of Vaccination (6.9%) | The regression revealed that family and religious leaders' attitudes were positively related to one's vaccination behavior and statistically significant.<br>While not among the top contributing drivers, healthcare providers' attitudes were significant, and were negatively related to vaccination behavior. |
| | $R^2 = .08***$ | | |
| **2: What drives people's intention to get vaccinated?** | i. Risk perception (17.1%)<br>ii. Ease of Vaccination (15%)<br>iii. Religious leaders' attitudes (12.6%)<br>iv. Own attitudes (12.2%)<br>v. Effectiveness (11.7%) | i. Family's attitudes (11.7%)<br>ii. Community's attitudes (11.1%)<br>iii. Healthcare providers' attitudes (8.6%) | The regression revealed that risk perception, effectiveness, ease of vaccination and religious leaders' attitudes were positively related to one's intention to get vaccinated and statistically significant. |
| | $R^2 = .67***$ | | |
| 3: What are the biggest and smallest self-ranked influences on people's decisions to get the vaccine? | For vaccinated respondents–<br>i. Own attitudes (43%)<br>ii. Ghana Health Services (25%)<br>iii. Family (11%)<br>For unvaccinated respondents–<br>i. Family (33%)<br>ii. Own attitudes (28%)<br>iii. Ghana Health Services (16%) | For vaccinated respondents–<br>i. Own attitudes (27%)<br>ii. Community (21%)<br>iii. Ghana Health Services (21%)<br>For unvaccinated respondents–<br>i. Community (27%)<br>ii. Family (25%)<br>iii. Own attitudes (20%) | Participants self-ranked who or what had the biggest and the smallest influence on their decision to get vaccinated.<br>In some cases the same variable showed up as both the biggest and smallest influence. For example, many participants ranked their own attitudes as having the biggest influence, while many others reported that their own attitudes had the smallest influence. |
| **4:** Which BI message type prompted people to recommend the COVID-19 vaccine to family and friends the most? | | | All BI message types were successful in prompting people to recommend the vaccine to friends and family, with 85–95% of the people agreeing to recommend across BI message types. |
| **5:** Which BI message type prompted people to share the benefits of the vaccine the most? | | | All BI message types were successful in prompting people to share the benefits of the vaccine, with 86–92% of the people agreeing to share across BI message types. |

*Note.* ***p < .001. Rescaled importance from the Relative Weights Analysis was used to identify the highest and lowest contributing drivers for research questions 1 and 2. The corresponding rescaled importance of each driver is indicated in parentheses for research questions 1 and 2.

room for improvement with respect to reducing barriers to vaccination at the time of data collection.

To test the influence of the eight potential drivers on respondents' vaccination behavior, a logistic regression analysis was conducted (Table 4). Specifically, we wanted to examine whether the proposed drivers significantly predicted vaccination behavior. The logistic regression model was statistically significant when examined using the Hosmer and Lemeshow goodness of fit test, $\chi^2(11, N = 1494) = 54.369$, $p < .001$. The model explained 7.56% (Cox and Snell Index $R^2$) of the variance in respondents' vaccination behavior. Of the eight drivers, family's attitudes and religious leaders' attitudes significantly predicted vaccination behavior in a positive direction; when respondents perceived family and religious leaders to have favorable attitudes towards the vaccine, they were more likely to get vaccinated. However, perceptions of

**Table 2. Descriptive statistics for vaccination behavior.**

|  |  | N | Vaccination Behavior (N = 1494) | | | |
|  |  |  | Yes | | No | |
|  |  |  | N | % | N | % |
| Overall |  | 1494 | 1089 | 73% | 405 | 27% |
| Gender | Male | 970 | 752 | 78% | 218 | 22% |
|  | Female | 524 | 337 | 64% | 187 | 36% |
| Location | Rural | 946 | 655 | 69% | 291 | 31% |
|  | Urban | 548 | 434 | 79% | 114 | 21% |
| Profession | Healthcare Workers | 298 | 256 | 86% | 42 | 14% |
|  | Non-healthcare Workers | 1196 | 833 | 70% | 363 | 30% |

healthcare providers' attitudes had a significant negative relationship with respondents' vaccination behavior. When respondents perceived healthcare providers to have positive attitudes towards the COVID-19 vaccine, they were less likely to get vaccinated. As shown in Table 4, one's own attitude, risk perception, effectiveness, ease of vaccination, and perceptions of the community's attitudes toward the COVID-19 vaccine did not significantly predict vaccination behaviors.

An important follow-up question pertains to whether the drivers included in the model varied with respect to their relative influence on vaccination behavior. Simply examining the size of regression coefficients cannot answer this question [40, 41]. Following the logistic regression analysis, a relative weights analysis (RWA) was therefore conducted to examine the relative importance/contribution of the drivers included in the regression model [41]. The relative importance of each predictor variable refers to the proportionate contribution that each predictor makes to the total predicted variance [42]. In addition to relative importance, RWA also provides confidence intervals, which can be used to evaluate both within and between sample differences in predictor weights [43, 44].

The results from any given RWA provide two estimates, the *raw importance*, and a *rescaled importance* for each predictor variable. The sum of raw importance of all predictors in a model result in the total model variance, $R^2$ [41]. It can be interpreted as the proportion of variance in the criterion variable (vaccination behavior) that is attributed to each of the predictor variables (drivers). The raw importance of predictor variables is each supplemented with confidence intervals, used to explain their precision [41]. Rescaled importance, on the other hand, is calculated by dividing the raw relative weight of each of the predictors in the model by the

**Table 3. Descriptive statistics of potential drivers for vaccination behavior: Vaccinated respondents.**

| Drivers | M | SD |
|---|---|---|
| **Risk perception** | 3.75 | 1.39 |
| **Effectiveness** | 3.80 | 1.26 |
| **Ease of Vaccination** | 3.66 | 1.35 |
| **Own attitudes** | 4.17 | 1.26 |
| **Family's attitudes** | 4.10 | 1.27 |
| **Community's attitudes** | 4.04 | 1.27 |
| **Religious leaders' attitudes** | 4.14 | 1.26 |
| **Healthcare providers' attitudes** | 4.21 | 1.21 |

*Note.* N = 1089

**Table 4. Results of the logistic regression analysis and relative weights analysis (RWA): Vaccination behavior.**

| Drivers | Logistic Regression | | | | Relative Weights Analysis | |
|---|---|---|---|---|---|---|
| | β | SE | z | p | Raw Importance | Rescaled Importance |
| Risk perception | -.072 | .072 | -.995 | .320 | .005 | 5.74% |
| Effectiveness | .042 | .070 | .603 | .547 | .008 | 9.67% |
| Ease of Vaccination | -.044 | .073 | -.606 | .544 | .006 | 6.86% |
| Own attitudes | .210 | .110 | 1.904 | .057 | .014 | 17.48% |
| Family's attitudes | .229 | .108 | 2.128 | .033* | .015 | 18.56% |
| Community's attitudes | .124 | .104 | 1.194 | .233 | .014 | 16.48% |
| Religious leaders' attitudes | .230 | .111 | 2.056 | .040* | .014 | 16.66% |
| Healthcare providers' attitudes | -.317 | .103 | -3.090 | .002** | .007 | 8.55% |
| $R^2$ | .0756 | | | | | |

*Note.* N = 1494

*$p < .05$

**$p < .01$. Rescaled importance (%) was calculated by dividing the raw weight by the model's total $R^2$ and then multiplying by 100.

total model variance, $R^2$ [41]. While raw importance uses the metric of relative effect sizes, rescaled importance uses the metric of percentage of the total predicted variance in a model that is attributed to each predictor variable [45].

Table 4 summarizes the RWA conducted to examine the relative importance of the eight drivers in the logistic regression model discussed above, with vaccination behavior as the criterion. Consistent with the regression model, family's attitudes and religious leaders' attitudes were amongst the highest raw and rescaled importance. Confidence intervals generated in the RWA output were used to test whether the eight drivers were significantly different from one another with respect to their influence. One's own attitudes and perceptions of their family's attitudes were each significantly more influential than risk perception, effectiveness and ease of vaccination. Perceptions of the attitudes held by the community, religious leaders and healthcare providers were also more influential than risk perceptions and ease of vaccination (see Table in Appendix B of S1 Appendix). When interpreting the role of healthcare providers' attitudes, it is important to keep in mind that this was a negative effect. Perceiving healthcare providers as supportive of the vaccine seemed to discourage vaccination behavior.

## Part I—Vaccination intention

Next, we examined what predicted intentions to get vaccinated among the 405 respondents who reported that they had not yet received the COVID-19 vaccine. Table 5 summarizes their

**Table 5. Descriptive statistics for unvaccinated respondents.**

| | | N | Vaccination Intention | |
|---|---|---|---|---|
| | | | M | SD |
| Overall | | 405 | 3.01 | 1.33 |
| Gender | Male | 218 | 3.15 | 1.38 |
| | Female | 187 | 2.84 | 1.26 |
| Location | Rural | 291 | 2.89 | 1.33 |
| | Urban | 114 | 3.32 | 1.31 |
| Profession | Healthcare Workers | 42 | 3.19 | 1.53 |
| | Non-healthcare Workers | 363 | 2.99 | 1.31 |

**Table 6. Descriptive statistics of potential drivers for vaccination intention: unvaccinated respondents.**

| Drivers | M | SD |
|---|---|---|
| Risk perception | 3.20 | 1.32 |
| Effectiveness | 3.21 | 1.34 |
| Ease of vaccination | 3.09 | 1.32 |
| Own attitudes | 3.38 | 1.40 |
| Family's attitudes | 3.30 | 1.39 |
| Community's attitudes | 3.26 | 1.40 |
| Religious leaders' attitudes | 3.38 | 1.38 |
| Healthcare providers' attitudes | 3.64 | 1.30 |

*Note. N* = 405

vaccination intentions overall, and by sociodemographic group. As can be seen, the overall vaccination intention hovered around the neutral midpoint of 3 and ranged from a low of 2.84 for women and a high of 3.19 for healthcare workers. As indicated by the standard deviations shown in Table 5, there was variability within each group with respect to intentions to get vaccinated. Table 6 shows descriptive statistics for each of the eight drivers as rated by the 405 respondents who reported not receiving the COVID-19 vaccine. As can be seen in Table 6, ratings ranged from a low of 3.09 to a high of 3.64. Specifically, the unvaccinated respondents perceived healthcare providers to have relatively positive attitudes toward the COVID-19 vaccine, with an average rating of 3.64. Respondents rated the ease of vaccination just above the midpoint, with an average of 3.09. Consistent with the average ratings previously reported for the vaccinated subsample, these results suggest that there was some scope for improvement in making it easier to get vaccinated at the time of data collection.

To test the influence of the eight drivers on respondents' vaccination intentions, a linear regression analysis was conducted (Table 7). The overall regression model explained 67.53% of the total variance in respondents' intention to get vaccinated ($R^2$ = .67, $F(8, 396)$ = 106, $p <$ .001). Of the eight drivers, risk perception, effectiveness, ease of vaccination, and religious leaders' attitudes significantly and positively predicted respondents' intention to get vaccinated. The influence of healthcare providers was again in the negative direction, but it was not statistically significant in the multiple regression analysis ($p$ = .147). Indeed, none of the four

**Table 7. Results of the linear regression analysis: Vaccination intention.**

| Drivers | β | SE | t | p | Raw Importance | Rescaled Importance |
|---|---|---|---|---|---|---|
| Risk perception | .279 | .045 | 6.159 | < .001*** | .117 | 17.13% |
| Effectiveness | .132 | .044 | 2.965 | .003** | .080 | 11.73% |
| Ease of Vaccination | .207 | .047 | 4.400 | < .001*** | .102 | 15.01% |
| Own attitudes | .118 | .068 | 1.746 | .081 | .083 | 12.19% |
| Family's attitudes | .089 | .074 | 1.201 | .230 | .080 | 11.69% |
| Community's attitudes | -.018 | .068 | -0.270 | .788 | .076 | 11.13% |
| Religious leaders' attitudes | .204 | .062 | 3.263 | .001** | .086 | 12.55% |
| Healthcare providers' attitudes | -.082 | .057 | -1.45 | .147 | .058 | 8.57% |
| $R^2$ | .6753 | | | | | |

*Note. N* = 405

*$p <$ .05; **$p <$ .01; ***$p <$ .001

remaining variables—own attitudes, family's attitudes, community's attitudes, and healthcare providers' attitudes—significantly predicted respondents' intention to get vaccinated.

The regression analysis was followed by an RWA to determine the degree to which the drivers in the model differentially influenced vaccination intentions. Results are shown in Table 7. Risk perception, ease of vaccination and religious leaders' attitudes had the highest raw and rescaled relative weights, highlighting their importance when it comes to vaccine intentions. Confidence intervals were used to test the differential importance of the eight drivers. When it comes to forming intentions to get the COVID-19 vaccine in Ghana, perceptions of healthcare practitioners' attitudes are significantly less important than nearly every other variable in the model: risk perception, ease of vaccination, own attitudes, family's attitudes, community's attitudes, and religious leaders' attitudes. No other significant differences among the drivers were found (see Table in Appendix C of S1 Appendix).

## Part I—Self ranking influences

The regressions and RWAs reported above reveal the influence of various factors on decisions and intentions to get vaccinated without asking people about who and what motivates them. This is a way to understand the kinds of unconscious drivers that affect human behavior on a daily basis. It is interesting to compare these results to people's conscious perceptions of who influences them when it comes to vaccinations. To explore this, we asked both vaccinated (N = 1089) and unvaccinated participants (N = 405) to self-rank which, among six sources (own attitudes, family, community members, Ghana Health Services, religious leaders and healthcare providers), had the biggest influence on their decision to get vaccinated and which among them had the smallest influence on their decision to get vaccinated. Table 8 summarizes the results.

Among vaccinated respondents, more than a third of the respondents ranked their own attitudes as most influential, indicating that they felt their own attitude toward the COVID-19 vaccine has the biggest effect on their decision to get vaccinated. At the same time, almost a quarter of the sample ranked this variable as having the smallest influence. The community was generally perceived to have a low level of influence, which is consistent with the regression analyses. Interestingly, a quarter of the respondents indicated that the Ghana Health Services has the biggest influence on their intention to get vaccinated and at the same time almost a quarter of the sample also ranked it as having the smallest influence on their intention to get vaccinated. Other self-reports were less consistent with the regression and relative weights analysis. For example, the results in Table 8 suggest that the influence of certain groups, such

**Table 8. Biggest and smallest self-ranked influence on vaccination intention.**

| Potential Influences | Vaccinated Respondents | | | | Unvaccinated Respondents | | | |
|---|---|---|---|---|---|---|---|---|
| | Biggest self-reported influence | | Smallest self-reported influence | | Biggest self-reported influence | | Smallest self-reported influence | |
| | N | % | N | % | N | % | N | % |
| **Own attitudes** | 465 | 43% | 292 | 27% | 114 | 28% | 83 | 20% |
| **Family** | 117 | 11% | 115 | 11% | 133 | 33% | 103 | 25% |
| **Community** | 27 | 2.5% | 254 | 23% | 18 | 4% | 109 | 27% |
| **Ghana Health Services** | 273 | 25% | 229 | 21% | 66 | 16% | 43 | 11% |
| **Religious leaders** | 106 | 10% | 98 | 9% | 34 | 8% | 35 | 9% |
| **Healthcare providers** | 101 | 9% | 101 | 9% | 40 | 10% | 32 | 8% |

*Note.* For vaccinated respondents, N = 1089; for unvaccinated respondents, N = 405

religious leaders, is largely unrealized by the study sample. While the models tested above (e.g., see Tables 4 and 7) clearly indicate that perceptions of what religious leaders think about the vaccine matter, respondents did not necessarily recognize this when asked to explicitly indicate who influences them.

Among unvaccinated respondents, a third of the sample indicated that family had the biggest influence on their intentions to get vaccinated, followed by a quarter of the sample that indicated that their own attitudes had the biggest influence. Both of these findings were not very consistent with the regression analysis. The community was again perceived to have a low level of influence, consistent with the regression and relative weights analysis. The impact of religious leaders' influence on respondents' intention to get vaccinated was quite low based on the self-rankings and inconsistent with the regression and relative weights analysis.

## Part II—BI message type (Jingle)

As discussed previously, in part II of the study, respondents were randomly assigned to one of the six BI message types shown in Appendix A in S1 Appendix. After listening to the assigned BI message, they were asked to report their willingness to recommend the COVID-19 vaccine to family members and friends. Table 9 shows the percentage of respondents exposed to each message type who expressed a willingness to recommend the COVID-19 vaccine. As can be seen, a large percentage of respondents, between 85–93%, expressed a willingness to recommend the vaccine to family members and friends with slight variations across BI message types. A chi-square goodness-of-fit test was performed to examine whether all six BI message types were equally effective in this regard. Results from a chi-square goodness-of-fit test suggested differential responses across the message types, $\chi^2$ (5, N = 1494) = 11.29, $p$ = .046 (Table 9). Pairwise comparisons with a Bonferroni correction were performed to determine whether some BI message types were more effective than the others in encouraging respondents to recommend the vaccine. Pairwise comparisons revealed that no one BI message type was more effective than the others.

Respondents also reported their willingness to share the benefits of the vaccine after listening to the assigned BI message type. Table 10 summarizes the results. Again, it appears all BI message types were quite effective, with 86–92% of the sample expressing a willingness to spread the word about the vaccine's benefits. A chi-square goodness-of-fit test was performed to examine whether all six BI message types generated similar responses. No significant differences among message types were indicated in the goodness-of-fit test, $\chi^2$ (5, N = 1494) = 8.42, $p$ = .135.

Table 9. Chi-square results for BI message type and willingness to recommend.

| BI Message Type | Willingness to recommend | | | |
|---|---|---|---|---|
| | Yes | | No | |
| | N | % | N | % |
| Fear | 212 | 89% | 25 | 11% |
| Altruism | 227 | 93% | 18 | 7% |
| Social norms | 214 | 88% | 29 | 12% |
| Ghana Health Services | 238 | 92% | 20 | 8% |
| Doctor | 209 | 85% | 36 | 15% |
| Religious leaders | 244 | 92% | 22 | 8% |

$\chi^2$ (5) = 11.29, $p$ = .046

*Note*. N = 1494

**Table 10. Chi-square results for BI message type and willingness to share.**

| BI Message Type | Willingness to share | | | |
|---|---|---|---|---|
| | Yes | | No | |
| | N | % | N | % |
| Fear | 215 | 91% | 22 | 9% |
| Altruism | 221 | 90% | 24 | 10% |
| Social norms | 209 | 86% | 34 | 14% |
| Ghana Health Services | 236 | 91% | 22 | 9% |
| Doctor | 213 | 87% | 32 | 13% |
| Religious leaders | 245 | 92% | 21 | 8% |

$\chi^2$ (5) = 8.42, $p$ = .135

*Note.* N = 1494

The effectiveness of the BI message types was similar for both vaccinated and unvaccinated respondents. Among unvaccinated respondents (n = 405), the effectiveness of the BI message types on respondents' willingness to recommend the vaccine, $\chi2$ (5, N = 405) = 9.35, $p$ = .095, and their willingness to share the benefits of the vaccine, $\chi2$ (5, N = 405) = 8.41, $p$ = .135, were not significantly differently from one another. Similarly, among vaccinated respondents (n = 1089), the effectiveness of the BI message types on respondents' willingness to recommend the vaccine, $\chi2$ (5, N = 1089) = 5.03, $p$ = .411, and their willingness to share the benefits of the vaccine, $\chi2$ (5, N = 1089) = 5.82, $p$ = .323, were not significantly differently from one another.

A final analysis was conducted on the unvaccinated subsample (N = 405), wherein, respondents once again reported their intentions to get vaccinated using the three items discussed previously. As shown in Table 11, vaccine intentions varied only slightly across message types, ranging between 3.26 and 3.47 on the 5-point Likert-type scale.

A one-way ANOVA was conducted to determine whether message type affected vaccine intentions. Results indicated no significant differences in respondents' intentions to get vaccinated based on the BI message they were assigned to, $F(5, 399)$ = 0.34, $p$ = .889 (Table 12).

## Discussion

The study provided important insights into factors that influence the COVID-19 vaccination behavior of the adult population of Ghana, the intention to take the vaccine among those who have not yet received it, as well as the efficacy of the various BI based audio message BI message types. The findings also indicate that a high number of community members are willing to

**Table 11. Vaccination intentions by BI message type.**

| BI Message Type | Vaccination Intentions | |
|---|---|---|
| | M | SD |
| Fear | 3.26 | 1.31 |
| Altruism | 3.28 | 1.33 |
| Social norms | 3.26 | 1.28 |
| Ghana Health Services | 3.39 | 1.28 |
| Doctor | 3.26 | 1.29 |
| Healthcare providers | 3.47 | 1.26 |

*Note.* N = 405

**Table 12. Effect of BI message type on vaccination intentions.**

|  | Sum of squares | df | Mean of squares | F | p |
|---|---|---|---|---|---|
| **BI Message Type** | 2.8 | 5 | .57 | .34 | .889 |
| **Residuals** | 667.3 | 399 | 1.67 |  |  |

*Note.* N = 405

share information and recommend vaccination to their friends, families, and neighbors. The number of willing volunteer vaccination promoters also demonstrate the importance of including questions in surveys not just about uptake and demand, but also about willingness to promote vaccination.

The study identified interesting differences in the intention to take the vaccine between vaccinated and unvaccinated respondents. Whereas a large proportion of the vaccinated respondents believed that the decisions to take the vaccine was influenced by their own attitude, a good number of unvaccinated believed that their intention was influenced by the community and family around. Accordingly, social influence, which is the process by which perceptions of what other people think and do influence beliefs and behaviors [46], should be considered carefully in COVID-19 vaccine demand generation strategies in Ghana. As social norms define what is acceptable in the given context [47], it is important to gain the trust of family members and religious leaders as advocates for COVID-19 vaccines. The importance of religious leaders in vaccine uptake is not surprising in the Ghanian context as respect for the traditional institutions and religious leaders is significant [48, 49]. Likewise, families play an important role in treatment and care for family members in Ghana [50, 51]. Although the influence of religious leaders for vaccine intention was not reported by many, religious leaders can be enlisted in community mobilization and engagement activities as they interact with diverse communities of Ghana, which include more than 70 ethnic groups, and they are seen as informal liaisons between local communities and state institutions. The role of religious leaders extends to political, economic, educational, religious, and family life, which makes them ideal messengers to family members [52–56]. Health authorities should build the capacity of religious leaders to become strong advocates for COVID-19 vaccines as well as to establish a network for social listening purposes to be able to assist religious leaders when community members have information voids and concerns regarding the vaccine. The Ghana Misinformation Task Force that was established at the beginning of the pandemic can be used as a structure to build the social listening mechanism [21]. As a good number of vaccinated and unvaccinated respondents also mentioned the influence of Ghana Health Services in their intention to take the vaccine, which highlights the importance to plan how to increase the visibility of them in vaccine demand creation.

The study found the perceptions of healthcare providers' positive attitudes towards COVID-19 vaccine having a significant negative relationship with respondents' vaccination behavior, though healthcare providers are often referred as a trusted and respected source of information among community members in Ghana [20, 57, 58]. Trust is known to be necessary for an effective vaccination program as the presence or absence of trust in patient-provider relationships in healthcare delivery can influence the provision of vaccine uptake [59]. Therefore, it is important to investigate further the association between healthcare providers' positive attitude and negative uptake of COVID-19 vaccines of community members. Trust building activities should be planned accordingly. Such efforts could focus on participatory approaches to ensure sustainability such as longitudinal stepwise interventions or simple interventions where the focus is improving communication and the goals of care [60].

Risk perception, effectiveness, ease of vaccination, and perceptions of the community's attitudes toward the COVID-19 vaccine did not significantly predict vaccination behaviors. However, risk perception and effectiveness positively predicted unvaccinated respondents' intention to get vaccinated and were statistically significant. In the literature, risk perception is widely recognized as a factor that motivates behaviors and practices including the uptake of COVID-19 vaccines [61, 62]. Likewise, in previous studies perceptions of the effectiveness of the COVID-19 vaccines have been associated with vaccine hesitancy in Ghana and elsewhere [63, 64].

The study also highlighted the usefulness of behavioral analytics that identified unconscious drivers affecting vaccine uptake and intentions by comparing it with respondents' conscious perceptions of factors influencing the uptake of the vaccine, which were varied. For example, the influence of religious leaders was rather unrealized whereas belief in the influence of their own attitude towards the vaccine was rather strong. BI offers a variety of interventions that can be used to influence behaviors and intentions of people. They can be used to address social constructs such as norms and altruism, as well as risk perceptions and efficacy perceptions. BI can also be used without trying to change what people think or feel through interventions such as reminders, defaults and other nudges. However more testing of various interventions is required [65].

All voice message based BI message types were followed by an overwhelming willingness of people to recommend the vaccine to family members and friends and to share COVID-19 related information with others. The message frames were also followed by favorable intentions to take the vaccine. The findings indicate that frames using fear, altruism, social norms, and health authorities, doctors and religious leaders as messengers can be used in vaccine demand creation in Ghana. Effective nudge-based messages have been identified and used successfully during the pandemic in other countries to encourage vaccine uptake [66] and to promote prevention measures such as social distancing [67, 68].

The current study demonstrates that the use of mobile phone-based surveys may prove to be an efficient methodology to reach people from rural areas. It is especially important since people from rural areas tend to be underrepresented in many studies examining behavior. Future research could focus specifically on identifying differences in the relative importance of drivers for vaccination behavior and intentions to get vaccinated between groups, such as, rural and urban areas.

The study had limitations. The sample may have been biased as 75% of the respondents reported having received at least one vaccination while according to GHS nationally only 22% of the population had received the first dose of the vaccine by end of January 2022. One explanation for this could be that the mobile phone survey database was based on people registered to receive health messages and that they may be more interested in health interventions including vaccinations than the population, on average, in the country. The sample may also have been biased because the majority of the respondents were young male from rural areas. It is therefore important to be careful when interpreting the findings. Generalizing the findings can be problematic. The study did not have a true control group for the BI message types, i.e., a message that did not include a BI nudge, nor a no-message control. This prevented a stronger test of the effects of the nudges embedded in the BI message types. An additional limitation was related to the cross-sectional study design that made it impossible to establish a causal relationship between factors of uptake and intentions of the COVID-19 vaccine. The findings of the study are expected to result in the development of interventions that aim to encourage uptake of COVID-19 vaccines. Future research should focus on measuring the impact of these interventions.

## Conclusion

The study identified factors that influence COVID-19 vaccine uptake including social influences by religious leaders and family, as well as factors that influence intention to take the vaccine among those who have not yet taken it, including risk perception, efficacy perception, the influence of religious leaders, and easy access to the vaccines. Effective message frames for COVID-19 vaccine promotion include fear, altruism, and social norms. Effective messengers for vaccine promotion include health authorities and religious leaders. The findings can be incorporated into COVID-19 vaccination strategies and plans.

## Supporting information

**S1 Table. Datafile.**
(XLSX)

**S1 Appendix.**
(DOCX)

## Acknowledgments

The authors would like to thank Jenna McChesney and Adam Meade for their contributions to this study.

## Author Contributions

**Conceptualization:** Swathi Vepachedu, Anastasiia Nurzenska, Anna-Leena Lohiniva, Al-hassan Hudi, Sena Deku, Julianne Birungi, Karen Greiner, Joseph Sherlock, Chelsi Campbell, Lori Foster.

**Formal analysis:** Swathi Vepachedu, Anastasiia Nurzenska.

**Funding acquisition:** Anastasiia Nurzenska.

**Investigation:** Lori Foster.

**Methodology:** Swathi Vepachedu, Anastasiia Nurzenska, Anna-Leena Lohiniva, Al-hassan Hudi, Sena Deku, Julianne Birungi, Karen Greiner, Lori Foster.

**Software:** Sena Deku.

**Supervision:** Anastasiia Nurzenska, Lori Foster.

**Validation:** Lori Foster.

**Writing – original draft:** Swathi Vepachedu, Anna-Leena Lohiniva.

**Writing – review & editing:** Anastasiia Nurzenska, Al-hassan Hudi, Sena Deku, Julianne Birungi, Karen Greiner, Joseph Sherlock, Chelsi Campbell, Lori Foster.

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
