## [Decision Letter · Decision Letter 0]

30 Mar 2023

PONE-D-22-26400Understanding COVID-19 vaccination behaviors and intentions in Ghana: A Behavioral Insights (BI) studyPLOS ONE

Dear Dr. Lohiniva AL,

Thank you for submitting your manuscript to PLOS ONE. After careful consideration, we feel that it has merit but does not fully meet PLOS ONE’s publication criteria as it currently stands. Therefore, we invite you to submit a revised version of the manuscript that addresses the points raised during the review process.

We look forward to receiving your revised manuscript.

Kind regards,

Mohan Kumar

Academic Editor

PLOS ONE

Reviewers' comments:

Reviewer's Responses to Questions

**Comments to the Author**

1. Is the manuscript technically sound, and do the data support the conclusions?

Reviewer #1: Yes

Reviewer #2: Yes

2. Has the statistical analysis been performed appropriately and rigorously? 

Reviewer #1: Yes

Reviewer #2: Yes

3. Have the authors made all data underlying the findings in their manuscript fully available?

Reviewer #1: Yes

Reviewer #2: Yes

4. Is the manuscript presented in an intelligible fashion and written in standard English?

Reviewer #1: Yes

Reviewer #2: Yes

5. Review Comments to the Author

Reviewer #1: Thank you for the opportunity to review this article detailing a mobile-phone based survey of Covid-19 vaccination behaviour and intention in Ghana. It was a pleasure to read this manuscript, it is a well-designed and well-conducted study, and it well written. I have the following comments for the authors:

Abstract: Include details of number (and proportion) of respondents who had received a Covid-19 vaccine versus unvaccinated respondents

Introduction

p.3 line 63 citation required for the assertion that vaccine hesitancy is a common issue with new vaccines

P.4 lines 94-101 it would be good to see further methodological details of these studies and outcome measures (I.e., vaccination behaviour or intention)

Methods

p.10 line 228 clarify here that only unvaccinated respondents were asked vaccination intention questions after the BI message

Power analysis suggests n = 2088 respondents required, but only analysed data from n = 1494 respondents, could the authors please comment on this, and whether it is consequential for the analyses and interpretation?

Effectiveness measure: consists of safety and effectiveness items, are these considered equivalent? One might think the vaccine is effective but not safe?

Cronbach’s alpha has been used as a measure of internal consistency for all relevant measures, but given the hight response to item ratio, a different measure of internal consistency may be more appropriate, e.g., IRT or McDonald’s Omega.

Results

For the analyses of self-ranking influences and BI message types, these should be stratified by vaccination status and conducted separately for vaccinated and unvaccinated respondents. I think it is a mistake to treat these two subgroups as drawn from the same population. The vaccinated and unvaccinated respondents may well report different rankings of influences on decision to get vaccinated. Similarly, the vaccinated respondents already hold pro-vaccination attitudes, and it would be interesting to see how the message type differs between vaccinated and unvaccinated respondents in terms of willingness to recommend the C19 vaccine and share the benefits of the vaccine.

Use consistent language and terms throughout, e.g., Table 1 and Table 8 variously talks about ‘highest/lowest contributing drivers’ and ‘biggest/smallest self-reported influence’ when reporting the results from the same items.

Discussion

If the authors re-analyse the data according to my recommendations above, the Discussion may need to change in places.

Discuss in more detail the significance of the highly vaccinated sample and the implications of the findings for the larger population; what gaps exist for promoting vaccinations in the unvaccinated population? Also, the conclusions are based on data from primarily young, male, rural dwelling respondents – can the authors discuss how this impacts/limits generalisability of findings?

Reviewer #2: Thank you for inviting me to review this paper. This is an interesting study aiming to identify factors associated with people’s COVID-19 vaccination status and their intention to take the vaccine in Ghana, as well as an experiment to test which of several behaviorally informed message frames had the greatest effect on vaccine acceptance. A few suggestions are listed below for the authors to consider for improvement of the paper.

Introduction

- There is no clear definition of vaccine hesitancy in the introduction. Please define it.

- If it is correct, BI appears for the first time on line 112 in the main text. Please give its full spelling when the abbreviation appears for the first time.

- Please give a clear description or definition of BI as this is a very important concept in this paper.

- Please give some examples of studies using BI to study health-related behaviors in the introduction. Doing so would give the readers a better about what BI is.

Method

- Line 151, 4 or 0.4?

- Some words were in italics. Is it necessary to do so due to the requirement of the journal? If not, please remove it.

Results

- Could the authors create a table showing the sociodemographic variables of the participants?

- Tables in academic papers are commonly presented in three-line tables. Please revise the tables.

Discussion

- One additional limitation of this study is that no causal relationship could be established between factors of the uptake/intentions of the COVID-19 vaccine since this is a cross-sectional study. Please add this in the limitation section.

Look forward to reading the revised version of this interesting paper.

6. PLOS authors have the option to publish the peer review history of their article (what does this mean?). If published, this will include your full peer review and any attached files.

Reviewer #1: No

Reviewer #2: No

---

## [Author Response · Author response to Decision Letter 0]

17 May 2023

We thank the editor and the reviewers for their comments that helped us to improve the manuscript. We have responded to each comment below. 

Reply: We have modified file name for supporting information to be S1 Table . 

We note that the grant information you provided in the ‘Funding Information’ and ‘Financial Disclosure’ sections do not match. When you resubmit, please ensure that you provide the correct grant numbers for the awards you received for your study in the ‘Funding Information’ section.

Reply: Funded by the UNICEF. 

Reply: Caption for S1 Table has been added in the end of the manuscript 

Reviewers' comments:

Reviewer's Responses to Questions

1. Is the manuscript technically sound, and do the data support the conclusions?

Reviewer #1: Yes

Reviewer #2: Yes

2. Has the statistical analysis been performed appropriately and rigorously?

Reviewer #1: Yes

Reviewer #2: Yes

3. Have the authors made all data underlying the findings in their manuscript fully available?

Reviewer #1: Yes

Reviewer #2: Yes

4. Is the manuscript presented in an intelligible fashion and written in standard English?

Reviewer #1: Yes

Reviewer #2: Yes

5. Review Comments to the Author

Reviewer #1: Thank you for the opportunity to review this article detailing a mobile-phone based survey of Covid-19 vaccination behaviour and intention in Ghana. It was a pleasure to read this manuscript, it is a well-designed and well-conducted study, and it well written. I have the following comments for the authors:

Abstract: Include details of number (and proportion) of respondents who had received a Covid-19 vaccine versus unvaccinated respondents

Reply: Thank you for suggesting this, we have now added it to the methodology section of the abstract, in the following language - 

“Data was collected from a total of 1494 participants; 1089 respondents (73%) reported already being vaccinated and 405 respondents (27%) reported not being vaccinated yet.”

Introduction

p.3 line 63 citation required for the assertion that vaccine hesitancy is a common issue with new vaccines.

Reply: Thank you for pointing out the need for a reference. We have added the following reference: Larson HJ, Gakidou E, Murray CJL. The Vaccine-Hesitant Moment. N Engl J Med. 2022 Jul 7;387(1):58-65. doi: 10.1056/NEJMra2106441. Epub 2022 Jun 29. PMID: 35767527;

P.4 lines 94-101 it would be good to see further methodological details of these studies and outcome measures (I.e., vaccination behaviour or intention). 

Reply: We have added more details about the studies including the outcome measures (intention, behavior) 

Methods

p.10 line 228 clarify here that only unvaccinated respondents were asked vaccination intention questions after the BI message

Reply: We have now clarified whether vaccinated, unvaccinated or both groups responded to the various sections of the survey. Throughout the procedure section, we indicated whether it was vaccinated, unvaccinated or all respondents that answered a question. For additional clarity, we have also indicated the corresponding sample size in the procedure section, p.10-12. 

For instance, line 244 on page 11 now reads, “All respondents (n=1494) then rated their perceptions on the various drivers of vaccination, namely, risk perceptions, ….”. Line 262 on page 11 now reads, “Unvaccinated respondents (n=405) were then once again presented with the three questions…”.

Power analysis suggests n = 2088 respondents required, but only analysed data from n = 1494 respondents, could the authors please comment on this, and whether it is consequential for the analyses and interpretation?

Reply: Thank you for pointing this out, while our power analysis does indicate that a sample size of 2088 would be ideal for general statistical analyses, a more specific test for a chi-square goodness of fit test, which was the statistical test used to detect differences in the experimental conditions, suggested that a sample size of 1494 is sufficient. 

Effectiveness measure: consists of safety and effectiveness items, are these considered equivalent? One might think the vaccine is effective but not safe?

Reply: The two items included as part of effectiveness measure have been adapted from the 5C scale developed by Betsch et al. (2018). According to this measure, perceptions of vaccine safety and effectiveness positively relate to confidence or trust in the vaccine and its effectiveness. With both of these items, we are attempting to measure respondents' behavioral beliefs about the vaccine , which would in turn relate to attitudes towards the vaccine. The two items were aggregated as suggested by Betsch et al. (2018) to create a composite. Further, the two items also have a reasonably strong internal consistency, as indicated using Cronbach’s alpha (α = 0.79). 

The Bestch et al. (2018) citation upon which this composite was based is as follows: Beyond confidence: Development of a measure assessing the 5C psychological antecedents of vaccination. PloS One, Betsch C, Schmid P, Heinemeier D, Korn L, Holtmann C, & Böhm R. 2018 Dec 7;13(12):e0208601-e0208601. https://doi.org/10.1371/journal.pone.0208601

Cronbach’s alpha has been used as a measure of internal consistency for all relevant measures, but given the hight response to item ratio, a different measure of internal consistency may be more appropriate, e.g., IRT or McDonald’s Omega.

Reply: Thank you for this comment. In an effort to provide additional clarification on the psychometric characteristics of these measures, we have now included a correlation matrix as a table in Appendix A and B for vaccinated and unvaccinated respondents (see below). Additionally, we have also reported McDonald’s omega along with the cronbach’s alpha in the measures section. 

Appendix A. Correlations between measures of drivers for participants who reported being vaccinated

Appendix B. Correlations between measures of drivers for participants who reported being unvaccinated.

Results

For the analyses of self-ranking influences and BI message types, these should be stratified by vaccination status and conducted separately for vaccinated and unvaccinated respondents. I think it is a mistake to treat these two subgroups as drawn from the same population. The vaccinated and unvaccinated respondents may well report different rankings of influences on decision to get vaccinated. Similarly, the vaccinated respondents already hold pro-vaccination attitudes, and it would be interesting to see how the message type differs between vaccinated and unvaccinated respondents in terms of willingness to recommend the C19 vaccine and share the benefits of the vaccine.

Reply:

We acknowledge and agree that it is important to distinguish between vaccinated and unvaccinated respondents when evaluating factors that were reported to have the biggest and the smallest influence on their vaccination intentions. We have now modified Table 8 (see below) to reflect this differentiation and also discuss the distinct findings for the two groups in the results sections. These findings are now also reflected in Table 1 to offer more clear insights into the factors influencing people’s vaccination intentions, both unvaccinated and vaccinated. 

Additionally, our results for the BI message types indicated very similar patterns in the effectiveness of BI message types on respondents’ willingness to recommend the vaccine and their willingness to share the benefits of the vaccine with others between individuals who were vaccinated and those that were not. Given these similarities, we have not reported findings by vaccination status. Chi-square test results also indicated that there were no significant differences in the effectiveness of BI message types for both vaccinated and unvaccinated respondents. Given these results, we have only included the results from the overall population, i.e., all respondents. However, we agree that it is relevant to address this and have now included a brief statement clarifying these results in the paper, which is as follows - 

“The effectiveness of the BI message types was similar for both vaccinated and unvaccinated respondents. Among unvaccinated respondents (n=405), the effectiveness of the BI message types on respondents’ willingness to recommend the vaccine, χ2 (5, N = 405) = 9.35, p = .095, and their willingness to share the benefits of the vaccine, χ2 (5, N = 405) = 8.41, p = .135, were not significantly differently from one another. Similarly, among vaccinated respondents (n=1089), the effectiveness of the BI message types on respondents’ willingness to recommend the vaccine, χ2 (5, N = 1089) = 5.03, p = .411, and their willingness to share the benefits of the vaccine, χ2 (5, N = 1089) = 5.82, p = .323, were not significantly differently from one another.”

Use consistent language and terms throughout, e.g., Table 1 and Table 8 variously talks about ‘highest/lowest contributing drivers’ and ‘biggest/smallest self-reported influence’ when reporting the results from the same items.

Reply: The language referring to the self-ranked influences has now been changed throughout the paper to be consistent. 

Discussion

If the authors re-analyse the data according to my recommendations above, the Discussion may need to change in places.

Reply: We have modified the discussion to highlight some differences between vaccinated and unvaccinated. 

Discuss in more detail the significance of the highly vaccinated sample and the implications of the findings for the larger population; what gaps exist for promoting vaccinations in the unvaccinated population? Also, the conclusions are based on data from primarily young, male, rural dwelling respondents – can the authors discuss how this impacts/limits generalisability of findings?

Reply: We agreed that it is important to point out the potential respondent bias. We have added in the study limitations the respondent bias and its implications to generalization of the results. We have included that in the limitations. The sample may have been biased as 75% of the respondents reported having received at least one vaccination while according to GHS nationally only 22% of the population had received the first dose of the vaccine by end of January 2022. One explanation for this could be that the mobile phone survey database was based on people registered to receive health messages and that they may be more interested in health interventions including vaccinations than the population, on average, in the country. The sample may also have been biased because the majority of the respondents were young male from rural areas. It is therefore important to be careful when interpreting the findings. 

Reviewer #2: Thank you for inviting me to review this paper. This is an interesting study aiming to identify factors associated with people’s COVID-19 vaccination status and their intention to take the vaccine in Ghana, as well as an experiment to test which of several behaviorally informed message frames had the greatest effect on vaccine acceptance. A few suggestions are listed below for the authors to consider for improvement of the paper.

Introduction

- There is no clear definition of vaccine hesitancy in the introduction. Please define it.

Reply: Thank you for pointing out the need for a definition. We have added it in the discussion The reference we used is as follows: MacDonald NE; SAGE Working Group on Vaccine Hesitancy. Vaccine hesitancy: Definition, scope and determinants. Vaccine. 2015 Aug 14;33(34):4161-4. doi: 10.1016/j.vaccine.2015.04.036.

- If it is correct, BI appears for the first time on line 112 in the main text. Please give its full spelling when the abbreviation appears for the first time.

Reply: We have corrected it. 

- Please give a clear description or definition of BI as this is a very important concept in this paper.

Reply: Thank you again for pointing out the need for a definition, We have added it in the discussion. Adhikari D. Exploring the differences between social and behavioral science. Behav Dev Bull. 2016;21(2):128–35. 10.1037/bdb0000029

- Please give some examples of studies using BI to study health-related behaviors in the introduction. Doing so would give the readers a better about what BI is.

Reply: We have explained how BI can be used in health related an in particura vaccine related behaviors px, lines x For example, BI uncover behavioral drivers by experiments and surveys using behavioral analytics to identify the relative influence and variety of factors that draw people to or away from vaccinations [28-30]. BI can also create strategies to improve decision-making, called nudges, which alter how choices are presented, leading decision-makers to behave in predictable ways. The key is to develop nudges in which the information is framed in a way that influences decision-making. COVID-19 message frame testing shows that different message frames work better in different environments, confirming the need for context specificity and highlighting the importance of identifying the appropriate frame for each context [31-33]. In addition, BI can encompass strategies that, unlike nudges, require sustained effort. For example, implementation scientists have long recognized that social motivation, incentives, and rewards are crucial levers of behavior change [34,35]. 

Method

- Line 151, 4 or 0.4?

Reply: 

Thank you for noticing this typo, we have now changed it to 0.4.

 - Some words were in italics. Is it necessary to do so due to the requirement of the journal? If not, please remove it.

Reply: We have removed italic from the words 

Results

- Could the authors create a table showing the sociodemographic variables of the participants?

Reply: Table 2 in the paper summarizes the demographic information collected from the participants of the survey, including their gender, whether they are from a rural or an urban location and whether they are healthcare or non-healthcare workers. Additionally, table 5 summarizes demographic variables specifically for unvaccinated respondents.

 - Tables in academic papers are commonly presented in three-line tables. Please revise the tables.

Reply: The format of the tables match PLOS One’s style guidelines. 

Discussion

- One additional limitation of this study is that no causal relationship could be established between factors of the uptake/intentions of the COVID-19 vaccine since this is a cross-sectional study. Please add this in the limitation section.

Reply: This has been added in the limitations bp x, lines x 

Look forward to reading the revised version of this interesting paper.

6. PLOS authors have the option to publish the peer review history of their article (what does this mean?). If published, this will include your full peer review and any attached files.

Do you want your identity to be public for this peer review? For information about this choice, including consent withdrawal, please see our Privacy Policy.

Reviewer #1: No

Reviewer #2: No

---

## [Decision Letter · Decision Letter 1]

19 Jun 2023

PONE-D-22-26400R1Understanding COVID-19 vaccination behaviors and intentions in Ghana: A Behavioral Insights (BI) studyPLOS ONE

Dear Dr. Lohiniva AL,

Thank you for submitting your manuscript to PLOS ONE. After careful consideration, we feel that it has merit but does not fully meet PLOS ONE’s publication criteria as it currently stands. Therefore, we invite you to submit a revised version of the manuscript that addresses the points raised during the review process.

We look forward to receiving your revised manuscript.

Kind regards,

Mohan Kumar

Academic Editor

PLOS ONE

Reviewers' comments:

Reviewer's Responses to Questions

**Comments to the Author**

1. If the authors have adequately addressed your comments raised in a previous round of review and you feel that this manuscript is now acceptable for publication, you may indicate that here to bypass the “Comments to the Author” section, enter your conflict of interest statement in the “Confidential to Editor” section, and submit your "Accept" recommendation.

Reviewer #1: All comments have been addressed

Reviewer #2: All comments have been addressed

2. Is the manuscript technically sound, and do the data support the conclusions?

Reviewer #1: Yes

Reviewer #2: Yes

3. Has the statistical analysis been performed appropriately and rigorously? 

Reviewer #1: Yes

Reviewer #2: Yes

4. Have the authors made all data underlying the findings in their manuscript fully available?

Reviewer #1: Yes

Reviewer #2: Yes

5. Is the manuscript presented in an intelligible fashion and written in standard English?

Reviewer #1: Yes

Reviewer #2: Yes

6. Review Comments to the Author

Reviewer #1: Thank you for the opportunity to review a revision of your article detailing a mobile phone based survey of Covid-19 vaccination behaviour and intention in Ghana. The authors have addressed all my comments, and I recommend publication of this important and interesting article.

Reviewer #2: (No Response)

7. PLOS authors have the option to publish the peer review history of their article (what does this mean?). If published, this will include your full peer review and any attached files.

Reviewer #1: No

Reviewer #2: No

---

## [Decision Letter · Decision Letter 2]

25 Sep 2023

Understanding COVID-19 vaccination behaviors and intentions in Ghana: A Behavioral Insights (BI) study

PONE-D-22-26400R2

Dear Dr. Lohiniva,

We’re pleased to inform you that your manuscript has been judged scientifically suitable for publication and will be formally accepted for publication once it meets all outstanding technical requirements.

Kind regards,

Khin Thet Wai, MBBS, MPH, MA

Academic Editor

PLOS ONE

Additional Editor Comments (optional):

All comments are fully addressed.

Reviewers' comments:

Reviewer's Responses to Questions

**Comments to the Author**

1. If the authors have adequately addressed your comments raised in a previous round of review and you feel that this manuscript is now acceptable for publication, you may indicate that here to bypass the “Comments to the Author” section, enter your conflict of interest statement in the “Confidential to Editor” section, and submit your "Accept" recommendation.

Reviewer #1: All comments have been addressed

Reviewer #2: All comments have been addressed

2. Is the manuscript technically sound, and do the data support the conclusions?

Reviewer #1: Yes

Reviewer #2: Yes

3. Has the statistical analysis been performed appropriately and rigorously? 

Reviewer #1: Yes

Reviewer #2: Yes

4. Have the authors made all data underlying the findings in their manuscript fully available?

Reviewer #1: Yes

Reviewer #2: Yes

5. Is the manuscript presented in an intelligible fashion and written in standard English?

Reviewer #1: Yes

Reviewer #2: Yes

6. Review Comments to the Author

Reviewer #1: Thank you for the opportunity to review a revision of your article detailing a mobile phone based survey of Covid-19 vaccination behaviour and intention in Ghana. The authors have addressed all my comments, and I recommend publication of this important and interesting article.

Reviewer #2: (No Response)

7. PLOS authors have the option to publish the peer review history of their article (what does this mean?). If published, this will include your full peer review and any attached files.

Reviewer #1: No

Reviewer #2: No

---

## [Editor Report · Acceptance letter]

3 Oct 2023

PONE-D-22-26400R2 

Understanding COVID-19 vaccination behaviors and intentions in Ghana: A Behavioral Insights (BI) study 

Dear Dr. Lohiniva:

I'm pleased to inform you that your manuscript has been deemed suitable for publication in PLOS ONE. Congratulations! Your manuscript is now with our production department. 

Kind regards, 

on behalf of

Dr. Khin Thet Wai 

Academic Editor

PLOS ONE